# Health and Nutrition Interventions to Prevent Childhood Overweight and Obesity in Mexico and Latin America: A Systematic Review

**DOI:** 10.3390/nu17243818

**Published:** 2025-12-05

**Authors:** Teresa Shamah-Levy, Marti Yareli Del Monte-Vega, Danae Gabriela Valenzuela-Bravo, Carmen Morales-Ruán, Lidia Moreno-Macías, Carlos Galindo-Gómez, Ileana Fajardo-Niquete, Javier Troconis-Cervera

**Affiliations:** 1Centro de Investigación en Evaluación y Encuestas, Instituto Nacional de Salud Pública, Cuernavaca 62100, Mexico; tshamah@insp.mx (T.S.-L.); ciee23@insp.mx (D.G.V.-B.); cmruan@insp.mx (C.M.-R.); 2Departamento de Nutrición Aplicada y Educación Nutricional, Instituto Nacional de Ciencias Médicas y Nutrición Salvador Zubirán, Mexico City 14080, Mexico; carlos.galindog@incmnsz.mx; 3Facultad de Medicina, Universidad Autónoma de Yucatán, Mérida 97000, Mexico; lmmacias@correo.uady.mx (L.M.-M.); ileana.fajardo@correo.uady.mx (I.F.-N.); 4Secretaría de Educación del Gobierno del Estado de Yucatán, Mérida 97130, Mexico; troloktc16@hotmail.com

**Keywords:** childhood obesity, school-age children, intervention study, systematic review, weight reduction

## Abstract

**Background:** Childhood obesity is a pressing global health challenge. Analyzing the efficacy of interventions is crucial to mitigate its impact and inform effective public health policies. This study aimed to conduct a systematic review of interventions (SRI) targeting school-aged children with obesity. Our goal was to identify the key components that contribute to the success of integrated interventions addressing diet/nutrition (D/N), physical activity (PA), and socioemotional skills. **Methods:** The Cochrane Collaboration methodology and the PRISMA statement were followed. The SRI included the following criteria, established a priori: studies that addressed obesity in school-aged children, including one or more interventions related to physical activity (PA), diet/nutrition (D/N), or socioemotional skills. Following the PICO (Population, Intervention, Comparison, and Outcome) framework, we searched six digital databases using relevant keywords and MeSH terms. The Mixed-Methods Appraisal Tool (MMAT) was used to assess article quality via “function group string” methods. Finally, a thematic synthesis of the SRI findings was conducted. The protocol for this study was registered in PROSPERO (CRD4202454214). **Results:** Initial screening yielded 127 articles. Following critical appraisal with the MMAT, studies with inadequate methodology, solely descriptive designs, unclear results, or interventions shorter than six months were excluded. Ultimately, 10 studies remained, eight of which included two of the three components of interest (D/N or PA). **Conclusions:** In this overview, many interventions were presented for the prevention of overweight and obesity in school-age children; however, methodological and standardized limitations still exist that hinder the establishment of effective interventions. Engaging families and teachers as active participants in interventions significantly enhanced effectiveness in both the D/N and PA domains. However, an analysis of current interventions highlights a stark gap in multisectoral and integrated approaches to tackling childhood obesity. This presents a remarkable opportunity for future initiatives to move beyond fragmented efforts and embrace a holistic model that unites families, schools, and communities to promote healthy lifestyles.

## 1. Introduction

Overweight and obesity (OW + Ob) are chronic multifactorial conditions with genetic, environmental, and lifestyle determinants that pose significant health risks, including type 2 diabetes, hypertension, dyslipidemia, and cardiovascular diseases [1,2]. The main factors related to OW+ Ob include physical inactivity and consumption of energy-dense foods [3], which present opportunities for prevention and treatment [4].

The prevalence of obesity in school-aged children has increased, mirroring the global trend, with the World Obesity Federation projecting a substantial increase by 2030 [5,6]. This surge is associated with nutritional transition [7,8], marked by the availability of low-cost processed foods, increased fast-food consumption, reduced home food preparation time, and heightened exposure to sedentary-inducing products [9,10]. In Latin America, particularly Mexico, Argentina, and Chile, 30% of those aged 5–19 years are affected [11].

Body image and parental perceptions significantly influence children’s access to care [12,13]. Early habits often persist into adulthood, underscoring the need to modify obesogenic environments for children’s long-term health [10,13]. Understanding the causal factors is crucial for addressing the high prevalence of obesity among school-aged children. Education promoting healthy habits, early weight gain detection, and targeted interventions is essential [14].

Public health interventions for childhood obesity vary, with single or multicomponent approaches targeting areas such as education, physical activity, and nutrition policy [9]. Previous reviews have focused on specific settings, such as homes or schools [15], yielding mixed findings. Recent systematic reviews have highlighted challenges in terms of evidence quality, intervention group comparisons, and study design clarity [15,16,17,18].

Additionally, recent meta-analyses have highlighted that while school-based interventions show small but significant benefits on BMI in children aged 5–11 years, these benefits may not be equally distributed across gender and socioeconomic groups. Evidence suggests that interventions tend to be more effective in boys, and there is limited information on whether they reduce health inequities, as measured by the PROGRESS framework [19].

A specific review in Mexico highlights the effectiveness of multicomponent interventions that combine physical activity, nutrition, and behavioral changes to prevent and reduce childhood obesity. The authors emphasized the methodological shortcomings and urged the scientific community to develop robust preventive and curative solutions [18].

Recent evidence has extended this body of work. An updated evidence report for the US Preventive Services Task Force, synthesizing 58 randomized clinical trials and more than 10,000 children and adolescents, found that multicomponent behavioral interventions are associated with small but meaningful reductions in BMI and other weight-related outcomes after 6–12 months, with larger effects in programs offering more hours of contact and structured physical activity sessions, although long-term effects on chronic disease and mortality remain unknown [20]. In addition, emerging trials published after our search period suggest that innovative modalities, such as a 10-week family focused e-health healthy lifestyle program [21], a multi-component school-based social network intervention that targets water intake and physical activity through peer influence [22], and a 6-month non-randomized trial in Norwegian children linking changes in diet, energy intake, and BMI *z*-score [23], may complement traditional school-based approaches, even though many of these interventions are short in duration and fall outside our inclusion criteria.

This systematic review aimed to summarize the essential elements in designing interventions for obesity in school-aged children, offering a critical evaluation to address knowledge gaps despite the high volume of available studies. By informing future researchers, this review aims to guide the development of effective and personalized programs for the Mexican school population.

## 2. Materials and Methods

This systematic review of interventions (SRI) adhered to the methodology proposed by the Cochrane Collaboration [24], and the PRISMA statement was followed with collaboration from trained personnel across five institutions. A team of two authors meticulously examined the compiled studies, assessing the risk and quality while standardizing the criteria and the fields of interest. Each author independently conducted a blind review of the scientific evidence and assessment methods.

The outcomes of this review serve as the cornerstone for the development of the Community Intervention of Integral Action for Children in Nutritious and Healthy Environments (INCAI in Spanish). The protocol for this review was approved by the Ethics, Biosafety, and Research Committees of the National Institute of Public Health in México, under Protocol CI:1791 1791 and registered in PROSPERO (CRD4202454214).

### 2.1. Search Strategy and Terms

A comprehensive search strategy was employed across six digital databases (ScienceDirect, DOAJ, Scielo, EbscoHost, Redalyc, and PubMed) using the PICO framework [25]. Collaboration with authors from five academic institutions provided diverse perspectives. Derived from Rivera Dommarco et al.’s book chapter [26], the search criteria covered themes such as obesity, school-aged population, program design, implementation, and evaluation. Key terms were adapted to the MeSH definitions, accounting for synonyms and MeSH Tree Structures.

Search chains utilizing key terms with a PICO structure and Boolean operators were constructed for the literature search. The inclusion criteria focused on community interventions aimed at enhancing health in local communities, guided by the definition of Trickett et al. [27]. The PICOS framework [25] directs the criteria, emphasizing the impact of community interventions on overweight and obesity prevalence in school-aged children.

The reviewers independently evaluated abstracts and titles, prioritizing original articles, controlled cohort studies, and longitudinal studies from Latin America until December 2023, with ongoing surveillance through January 2025. The minimum intervention duration of six months was selected a priori to focus on programs with sufficient exposure time to influence medium-term anthropometric and behavioral outcomes of the participants. Consequently, shorter pilot or feasibility studies, including some school-based programs, were excluded from this review.

The outcome measures included a reduction in the prevalence of overweight or obesity, BMI *z*-score reduction, improved dietary habits, adoption of healthy habits, and increased physical activity. This meticulous approach ensured a comprehensive and diverse exploration of interventions for childhood obesity in the school context.

### 2.2. Data Extraction

Predefined data extraction forms were used to record the study characteristics. Each author was responsible for completing the forms, and the two authors/researchers reviewed and compared the extracts containing the following information:
General information: database; search string; PICO elements; Digital Object Identifier (DOI); author; year of publication; title; country; study design and type; language; identification of control or intervention group; total sample size; population evaluated.Intervention details: diagnostic criteria for overweight or obesity, type of intervention (nutritional, physical activity, educational, or mixed), study summary, duration of intervention (in months), intervention site, population included in the intervention, and intervention staff.Outcomes: Changes related to the program, such as reduction in obesity components, lifestyle changes, dietary changes, and changes in physical activity.The two reviewers independently completed the extraction forms and compared them. Any discrepancies in study eligibility or extracted data were discussed until a consensus was reached before the data were entered into the final database.Results: Program-related changes, such as a reduction in obesity components, lifestyle changes, dietary changes, and physical activity changes, were observed.

### 2.3. Strategy for Selection and Assessment

After compiling and standardizing the articles using extraction forms, a thorough assessment was conducted using IBM SPSS Statistics V22.0 with group string functions and simple descriptive analyses in the exploratory review.

Dictionaries were created to standardize variable definitions, including journal names, DOI, country, intervention duration, and evaluated population. A code was developed to associate each country with its region using an international classification system. Mexico was classified separately because of its significant relevance, and the other classifications included North America, Latin America, the Caribbean, Asia Pacific, Europe, the Middle East, North Africa, and Southern Africa [28].

For critical assessment, the Mixed-Methods Appraisal Tool (MMAT) version 18 was employed [29], utilizing a two-phase scoring system involving base filtering and specific filtering based on the study methodology. The criteria included screening/diagnostic and quantitative methods (controlled and non-randomized) [30].

Although we did not calculate formal inter-rater reliability statistics, the use of independent dual screening and consensus procedures followed the current Cochrane Collaboration guidelines for minimizing selection and assessment bias.

Studies that met the inclusion criteria and passed the methodological assessments were independently reviewed by two authors. Key participant and intervention characteristics were summarized using standard templates, and narrative synthesis was employed to report data on favorable and unfavorable outcomes. This meticulous approach ensured a comprehensive and robust evaluation of the selected studies on childhood obesity interventions in the school context.

We adhered to the PRISMA guidelines for the development and reporting of systematic reviews. The process of study identification, screening, eligibility, and inclusion was conducted following the methodological recommendations established by PRISMA 2020 [31] (Appendix A).

## 3. Results

### 3.1. Description of Selected Studies

A total of 127 articles were initially collected, and after excluding 30 duplicates and articles without a DOI, 97 articles remained (Figure 1) [31]. The first exploration eliminated studies with a transversal or descriptive design and those describing the study protocols, resulting in a final set of 59 articles for the analysis. Articles published in English or Spanish within the past 11 years were selected for this systematic review.

Articles whose descriptions were unclear, lacking critical methodological aspects, or involving interventions of less than six months were discarded. Following database cleaning, additional filters were applied to identify the relevant characteristics for this study. Two researchers independently and blindly evaluated the methodological quality of the 10 included studies using the Mixed-Methods Appraisal Tool (MMAT 2018 version), resolving discrepancies by consensus. The included studies corresponded to longitudinal, intervention, and quasi-experimental designs, all of which had moderate to high methodological quality according to the MMAT criteria. The main identified limitations were the absence of blindness for participants and evaluators (justified by the nature of the behavioral interventions in the school-age population), loss of follow-up, or difficulty in obtaining complete measurements due to deficient selection processes. These limitations were considered in the interpretation of the results and in future research recommendations.

Variables such as intervention type, geographic location, language, and publication year were examined. Most studies were conducted in North America, Europe, Latin America, the Caribbean, Asia–Pacific, and Mexico. The selected studies described interventions with a median population of 2124 participants and a median duration of 17.8 months.

This review focused on articles that specifically addressed obesity in school-aged populations, including one or more interventions (physical activity, diet/nutrition, and socioemotional skills), conducted in the school environment and included community-based components.

The primary measures in the 10 selected studies included a reduction in the prevalence of overweight and obesity, reduction in the *z*-score of BMI for age, improvement in dietary habits, promotion of healthy habits, and increase in physical activity. Eight studies were multicomponent, addressing both diet/nutrition and physical activity, while two focused solely on diet/nutrition without a control group. Three studies considered socioeconomic status (SES) and indigenous or minority groups. The median study duration was 22 months, ranging from 8–43 months). Four studies specifically considered SES, whereas two did not mention any specified characteristics.

Anthropometric measurements, particularly height and weight for estimating BMI and nutritional status, were used in all 10 studies, with half utilizing reference values from the World Health Organization. Six studies included additional risk components, such as waist circumference, and only one study reported the measurement of biochemical markers to detect metabolic syndrome (Table 1).

Owing to the specific inclusion criteria prioritizing studies in Mexico and Latin America and the focus on multi-component interventions, the limited number of studies obtained prevented the establishment of a meta-analysis. The heterogeneous results, combined with the lack of methodological rigor in some studies, may compromise the interpretation or lead to unreliable and inconsistent statistical synthesis.

Consequently, we intentionally chose a narrative and thematic synthesis approach, focusing on patterns in intervention components, settings and implementation strategies rather than on pooled quantitative estimates of effect.

### 3.2. Diet and Nutrition

The identified interventions predominantly used educational lessons, workshops, and sports clubs, focusing on diet/nutrition (D/N). Nine of the ten interventions were implemented within schools, four of which included a consistent nutritionist presence. Health sector personnel included physicians, nurses, nutrition educators, and health professionals.

Parental involvement was a significant feature of the eight interventions with D/N components, and only two were extended to the broader community. The effects included increased demand for healthier foods in school cafeterias, improved food availability, changes in unhealthy food consumption, increased vegetable consumption, and reduced consumption of high-fat snacks. Tarro [38], Bacardi [32], and Waters [40] focused on healthy food selection education, increased fruit and vegetable consumption, and decreased consumption of sugary and fatty snacks. Others, such as Safdie [36] and Long [34], have enhanced the school food environment by increasing the availability of healthy foods, reducing the availability of high-calorie foods and sugary beverages, and incorporating nutrition education.

Some studies, such as the Kids Nutrition and Fitness (KNF) program [42], provided extracurricular nutrition education, while Vazquez [39] and Scruzzi [37] focused on nutrition education through workshops, evaluating the impact on consumption before, during, and after the intervention. Parental involvement and school environment modifications have emerged as promising aspects for success in reducing childhood obesity.

Flynn’s study [43] underscored the significance of home interventions and family involvement in preventing obesity in children under five. The active engagement of adults, especially parents and teachers, is crucial for favorable outcomes, such as weight reduction and obesity prevention. However, Morgan et al. [41] suggested the need for additional evidence, emphasizing the importance of primary studies detailing intervention designs and outcome measures to establish the value of caregiver involvement.

### 3.3. Physical Activity

In interventions incorporating physical activity components, physical education teachers played a central role in five of eight cases, with classroom teachers providing support. Only two interventions involved a leading health professional. Four interventions demonstrated positive effects on physical activity levels among school-aged children, emphasizing strength-building activities, additional physical education sessions, and improvements in sports and playground equipment.

All studies focused on interventions during regular school hours, with only one extending the activity to the home. Five studies involved parents, and two showed significant changes, increasing physical activity hours and reducing sedentary time, compared to the control groups.

In a review by Liu et al. [44], BMI variations were noted with physical activity; however, no significant differences were found between interventions with multiple components and those focusing on a single component. Ameryoun et al. [45] found that game-based interventions had a consistent but small effect on reducing BMI *z*-scores, especially when combined with nutrition and education.

Bleich et al.’s review [46] highlighted that prioritizing physical activity sometimes reduced interventions in education and dietary components. Synthesizing the evidence emphasizes the importance of various activities in obesity reduction programs. School-based interventions, particularly strength sports and active games, offer promising opportunities for study. Improving the evidence in these areas can help identify effective strategies for reducing excess weight in early childhood.

### 3.4. Overweight and Obesity

Several studies, including those by Sadeghi [47], Bacardi [32], Tarro [38], and Vazquez [39], demonstrated significant reductions in obesity measures among school-aged children in their interventions compared to the control groups. Safdie’s [36] basic intervention showed the greatest reduction in the prevalence of overweight or obesity, which dropped from 12.1% to 10.9%. Bacardi-Gascon noted a primary change in abdominal obesity percentage, decreasing from 20.6% to 15.2% [32].

However, Waters [40] and Centeio [33] reported decreased overweight and obesity proportions in both the intervention and control groups, complicating the attribution of results solely to the interventions. Brown et al. [48] review suggested that combined physical activity and diet interventions in preschoolers reduce obesity risk and BMI-for-age *z*-score. In 6–12-year-olds, physical activity interventions reduced BMI, *z*-score, and overweight/obesity prevalence, but the difference was not statistically significant. Dietary interventions in this age group lacked conclusive evidence, whereas combined interventions reduced the BMI *z*-score for age.

Cauchi et al. [49] reported small-to-modest impacts on anthropometric outcomes and childhood obesity reduction in most interventions, with no clear link between intervention components and effectiveness. A notable limitation of this review and others is the lack of methodological clarity, especially in statistical power calculations, which hinders the precise interpretation of observed variations. Clearer reporting of methodological criteria is essential for a more accurate understanding of effective approaches to address childhood obesity.

### 3.5. Socioemotional Skills

Few studies have emphasized socioemotional skills and behavioral changes as success factors in treating childhood overweight and obesity. Giannin et al. [47] found that caregivers enrolled children for physical and mental health improvement, especially when medical issues, such as high blood pressure and probable diabetes, were identified. The caregivers acknowledged that small changes at home led to weight loss and habit changes, acting as a catalyst for further positive changes.

Waters et al. [40] and Vásquez et al. [39] integrated psychological aspects into their interventions. Waters included self-esteem promotion, whereas Vasquez included sessions with a psychologist to support behavioral changes related to diet and physical activity. These findings highlight the significance of addressing not only physical health but also socioemotional well-being and behavioral modifications in effective childhood obesity interventions.

## 4. Discussion

Our analysis emphasizes the crucial role of intervention time in addressing childhood obesity and emphasizes its impact on program design and implementation. Among the studies analyzed, nine interventions occurred within the school environment during regular hours, five extended hours, and two involved community components.

Dividing treatments for overweight and obesity, as seen in other reviews, helps reduce result heterogeneity and prioritizes recommendations based on effectiveness. The review by Long et al. [34] highlights that a long intervention time does not guarantee effective adherence or reduction in obesity, suggesting the need for modifications when transitioning from short-term to long-term interventions.

Effective interventions lasting between eight and nine months have proved to be successful in addressing childhood obesity. Strategies combining proper nutrition, promotion of fruit and vegetable consumption, water intake, and physical activity in schools over 4 to 12 months showed positive effects on maintaining or slightly improving BMI. However, standardizing the design and implementation remains as a common need across studies.

Our findings align with global evidence that the effectiveness of obesity prevention interventions may differ across subgroups. Palmer et al. [19] in 2025 showed in a re-analysis of 81 trials, interventions had a greater impact on boys aged 5–11 years and did not significantly affect inequities across other PROGRESS domains. These insights support the need for equity-sensitive designs in future interventions, especially in Latin American contexts where social and economic disparities are profound.

Consistent with these results, the updated JAMA evidence report for the US Preventive Services Task Force reported that behavioral weight-management interventions for children and adolescents yield modest average reductions in BMI at 6–12 months, with larger effects observed in programs with higher contact time and supervised physical activity sessions, while highlighting the lack of long-term randomized evidence on cardiometabolic outcomes and mortality [19]. Taken together, Palmer et al.’s [19] re-analysis and the US Preventive Services Task Force report underscore the importance of designing equity-sensitive, long-term interventions that monitor effects across PROGRESS factors, rather than relying solely on short-term changes in BMI.

In Mexico, the “Nutrition on the Go” strategy, which focuses on reducing the energy content of school breakfasts, including fruits and vegetables, and promoting physical activity and water consumption over six months, had a statistically significant effect on reducing the probability of shifting from overweight to obesity categories. This study also documented the decreasing effect of the shift from the normal to overweight categories during the intervention [50].

Beyond the national “Nutrition on the Go” strategy, several school-based interventions conducted in Mexico and South America provide additional regional context for our findings.

In Mexico, Bacardí-Gascon et al. [32] reported primary school interventions that combined nutrition education with changes in the school food environment to prevent obesity, while Safdie et al. [36] evaluated a comprehensive program that integrated healthier food offerings, promotion of physical activity, and educational components to reduce obesity risk factors.

In Chile, Vásquez et al. [39] examined a secondary prevention program among children with obesity that emphasized lifestyle modification within the school setting. Together, these Latin American experiences show that multicomponent school- and community-based strategies are already being implemented and are consistent with broader evidence that childhood obesity in the region is driven by interacting structural determinants, including unhealthy food environments, social inequity and political–economic interests [51], as well as marked socioeconomic and ethnic gradients in overweight and obesity among Mexican and Indigenous children and adolescents [52,53].

Complementary global work in high-risk minority populations indicates that culturally contextualized lifestyle interventions that integrate diet and physical activity and actively engage families and communities can improve adiposity and cardiometabolic risk [54], reinforcing the need for equity-oriented intersectoral school- and community-based programs in Mexico.

Long et al.’s study [34], emphasizing that a longer intervention (36 months) did not guarantee effective results, recognized the limited impact on BMI, potentially due to the extension of a short-term intervention without necessary modifications. In summary, intervention length is a critical factor, and careful consideration and adaptation are essential for effective and sustained outcomes in childhood obesity programs.

Interventions involving families and teachers had a greater effect on both diet and physical activity, contributing to positive outcomes. This highlights the importance of collaborative programs involving multiple stakeholders.

Although our inclusion criteria focused on Latin American, particularly Mexican, community- and school-based interventions lasting at least six months, recent shorter or non–Latin American trials can also provide information about future research agendas. Family-focused e-health programs have demonstrated reductions in BMI *z*-score over a 10-week period among school-aged children with overweight or obesity [20], and a school-based social network intervention has shown that peer influence can be leveraged to promote water consumption and physical activity and to improve BMI trends, even without direct parental involvement [21]. Moreover, a 6-month non-randomized trial in Norwegian children illustrated that changes in home food availability, total energy intake, sugars, and saturated fat are closely related to reductions in BMI *z*-score [22]. Building on these findings, future Latin American interventions should test hybrid models that combine longer school- and community-based components with digitally delivered, family-centered strategies; explicitly target the home food environment and beverage consumption; and prospectively plan subgroup analyses by sex, socioeconomic status, and other PROGRESS characteristics, to ensure that new programs reduce—rather than exacerbate—existing health inequities.

Addressing the Latino and/or Mexican population in the United States is increasingly crucial due to their heightened vulnerability to obesity, which is often tied to their social and economic environments.

Various techniques and theories, such as socio-ecological theory, the theory of social education, and ecological methods for healthy habit formation, have been identified in interventions. This was particularly prevalent in programs aimed at instilling habits through education.

Focusing on the school-aged population, specifically targeting ages eight to nine, highlights the segmentation of physical and learning activities. In the Mexican context, this corresponds to first-to third-year students in the first group and fourth-to sixth-year students in the second group. However, the inclusion of different grades and ages in the assessed studies may limit the generalizability of their findings. Overall, these insights emphasize the importance of tailored approaches, collaboration, and cultural considerations when addressing childhood obesity.

Our study has several limitations. Only 10 of the 127 identified articles met all inclusion and quality criteria, as we restricted the review to multi-component community- and school-based interventions of at least six months conducted mainly in Mexico and Latin America. This improved internal coherence but reduced representativeness and excluded shorter pilot or feasibility studies. Substantial heterogeneity in intervention components, settings, comparison groups, and outcomes, together with the absence of a control group in two studies and variability in methodological quality, precluded meta-analysis and limited the precision of effect estimates; therefore, we relied on thematic and narrative syntheses. In addition, contextual factors such as socioeconomic, cultural, and environmental conditions were not consistently measured or analyzed, despite three studies reporting socioeconomic status or membership of indigenous or minority groups, which constrains the generalizability of our findings across population subgroups. Finally, although we used the MMAT to appraise methodological quality in a structured way, we did not calculate inter-rater reliability, and future reviews should report screening agreement and risk-of-bias assessments in greater detail.

## 5. Conclusions

In this overview, plenty of interventions were presented for the prevention of overweight and obesity in school-age children. Interventions aimed at preventing, controlling, or reducing obesity generally lack a multisectoral approach and integrated components. Providing dietary and nutritional education to the entire population is crucial for reducing the prevalence of non-communicable diseases by promoting healthy habits and behaviors.

The implementation of interventions requires collaboration between the health and education sectors, considering the influence of the social determinants of health. Recognizing and addressing obesogenic environments is particularly important. These considerations emphasize the need for methodological rigor, multisectoral approaches, and a comprehensive understanding of social determinants in designing effective interventions for childhood obesity.

Intervention policies intended to prevent obesity in schoolchildren should consider changes within the school environment itself through actions such as improving physical education classes and creating more implementable nutritional policies. For a greater impact, the inclusion of parents and teachers—in addition to government officials, communities, and civil society as elements of healthy lifestyles, as well as the development and well-being of society—is recommended.

For Mexico and other Latin American countries, our synthesis highlights the potential of multicomponent school and community-based interventions that integrate nutrition, physical activity, and socioemotional components tailored to local sociocultural contexts to inform regional policies and large-scale implementation.

Among the fundamental characteristics that must be included in public policies are registry and screening systems that allow for timely diagnostics to direct adequate interventions, as well as flexibility in response to several situations that consider cultural diversity and school environment scenarios. Currently, public policies in Mexico and Latin America show disadvantages, such as the interruption of programs as a consequence of government transitions, limited transparency in the results obtained, and the weak articulation between actions and sectors involved in their implementation. A successful public policy for obesity and overweight prevention must be capable of identifying the changes attributable to the program or directed strategy, as well as perform the pertinent adjustments and guarantee the long-term sustainability of concrete actions.

## Figures and Tables

**Figure 1 nutrients-17-03818-f001:**
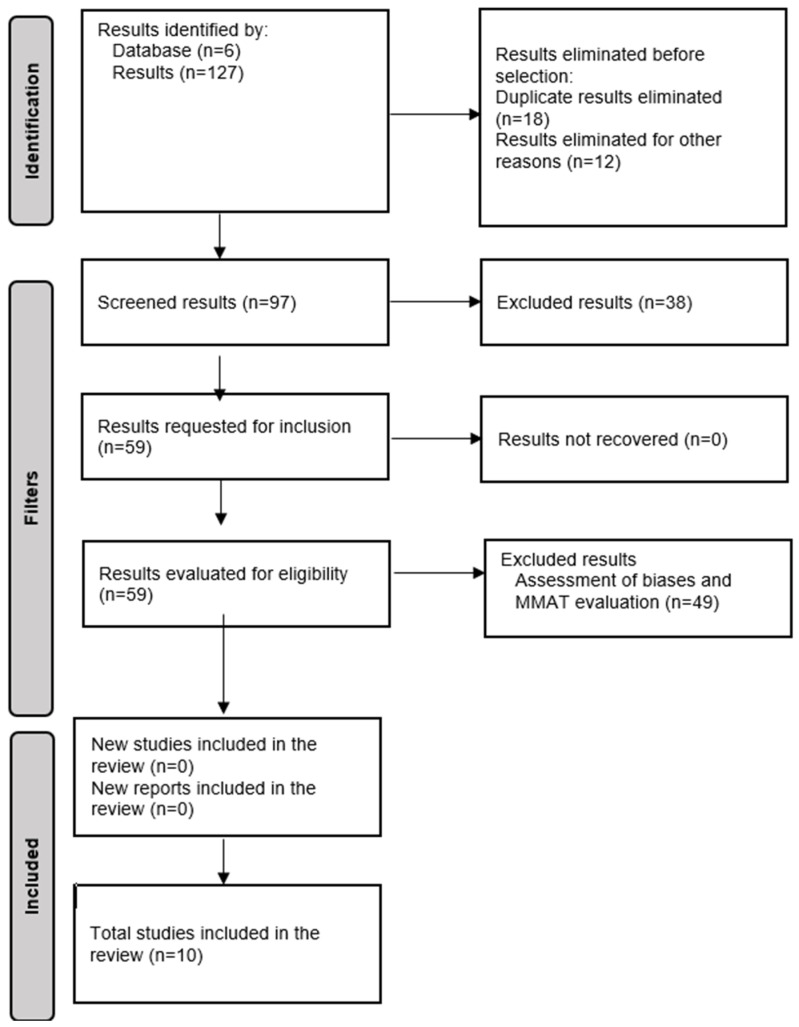
PRISMA Flowchart [31].

**Table 1 nutrients-17-03818-t001:** Key characteristics of the studies.

Objective	Summary Highlights	Intervention Length (Months)	Country	Participants (Intervention = I, Control = C)	Implementation Area for Dietary, Nutritional, and Physical Activity Components	Measurements Used to Estimate Obesity
A six-month randomized school intervention and an 18-month follow-up intervention to prevent childhood obesity in Mexican elementary schools [32].						
To assess the effect of a six-month intervention and an 18-month follow-up intervention on the body mass index, food consumption, and physical activity of second- and third-grade elementary school children.	A randomized trial with 532 participants showed a significant BMI reduction of −0.82 (*p* = 0.0001) at six months. At 24 months, there were notable increases in the BMI *z*-score and waist circumference, with reduced abdominal obesity, 8% remission, and 18% overweight/obesity incidence. Comprehensive intervention increased vegetable intake (*p* = 0.007) and physical activity (*p* = 0.0001) while reducing sedentary behaviors and high fat/salty snack consumption.	24	Mexico	I: 280, C: 252	School setting in regular school hours, school setting extracurricular	Weight, Height, BMI, Waist Circumference
Building healthy communities: A comprehensive school health program to prevent obesity in elementary schools [33].						
This study examined the impact of a socioecological theory driven by school-wide nutrition and physical activity interventions on 5th graders’ central adiposity and obesity levels.	A school-wide program integrating nutrition and physical activity, guided by socioecological theory, reduced central adiposity in 5th graders. Over eight months, 377 treated students saw significant decreases in central adiposity compared to 251 controls, hinting at potential BMI reductions. This finding highlights the effectiveness of holistic school interventions in promoting children’s health.	8	United States of America	I: 2019, C: 1035	School setting in regular school hours, school setting extracurricular	Weight, Height, BMI, Waist Circumference, Waist-Hip Ratio
Evaluation of a pragmatic trial of a collaborative school-based obesity prevention intervention in a low-income urban district [34].						
To evaluate changes in diet and physical activity in intervention schools using surveys and direct observation	This study explored whether empowering school leaders to choose suitable components from evidence-based programs could overcome barriers to their implementation. A customized intervention was applied from 2013 to 2016 in nine schools across a low-income urban district. While the program demonstrably lowered the BMI of students in the two semesters, the effect did not persist. Despite this, the study proved a valuable method: it empowered school leaders, leveraged existing district resources, and reduced the evaluation burden. These insights offer a promising avenue for future obesity prevention.	36	United States of America	I: 3152, C: 2730	School setting in regular school hours, school setting extracurricular and summer camp	Weight, Height, BMI
A three-year multifaceted intervention to prevent obesity in children of Mexican heritage [35].						
To estimate the effects of a multifaceted, community-based intervention on body mass index (BMI) among Mexican heritage children.	This study evaluated the effects of the Niños Sanos and Familia Sana interventions on BMI among Mexican-heritage children in California’s Central Valley. Over three years, the multifaceted program included parent workshops, school nutrition lessons, enhanced physical education, and monthly fruit and vegetable vouchers. It significantly slowed BMI growth in boys and reduced BMI in obese girls, underscoring the need to address gender disparities and collaborate with stakeholders to combat childhood obesity.	36	United States of America	I: 387, C: 313	School setting in regular school hours, school setting extracurricular, community involvement	Weight, Height, BMI, Waist Circumference
Impact of a school-based intervention program on obesity risk factors in Mexican children [36].						
To evaluate the impact of an 18-month intervention to prevent obesity in fourth- and fifth-grade students grounded on the ecological model of healthy behaviors in Mexico.	This study was guided by ecological principles and formal research. Employing a randomized controlled trial (RCT) design, 27 schools were assigned to basic or plus interventions and control conditions in a sample of 830 students. Over two years, data on the school environment, children’s eating and physical activity, and BMI were collected from 830 students. Intervention schools showed an increased availability of healthy foods and decreased unhealthy options, with a similar trend in food intake. No significant increase in moderate-to-vigorous physical activity (MVPA) during PE or recess was observed; however, an increase in steps taken was noted.	18	Mexico	I: 526 + 262	School setting in regular school hours	Weight, Height, BMI
School Health: An Educational Nutrition Intervention from a Comprehensive Approach [37].						
To encourage healthy lifestyles among school-age children attending a public school in Córdoba, Argentina (2013–2014).	An interdisciplinary intervention (2013–2014) aimed to promote healthy lifestyles through diagnoses, food education, and healthy kiosks. The findings highlight concerns such as being overweight, sugary drink consumption, and school food kiosks. The project successfully achieved its goals by emphasizing the importance of family and school communities in shaping healthy eating habits.	12	Argentina	I: 127, C: N/A	School setting in regular school hours, school setting extracurricular, did not include a physical activity component	Weight, Height, BMI
A primary-school-based study to reduce the prevalence of childhood obesity--the EdAl (Educacio en Alimentacio) study: a randomized controlled trial [38].						
The aims of the study were: 1) to design a health promotion program for implementation by HPAs in primary schools, and 2) to evaluate the effects of a 3-year school-based program of lifestyle improvement, including diet and physical activity over a period of 28 months, on the prevalence of obesity.	In a randomized comparison, the intervention group showed a 2% reduction in obesity prevalence among boys, resulting in a significant drop in BMI *z*-scores. More intervention boys achieved the recommended physical activity levels, with increased fish consumption protecting against obesity, emphasizing the effectiveness of university student-led interventions in reducing childhood obesity.	24	España	I: 1222, C: 717	School setting in regular school hours, school setting extracurricular	Weight, Height, BMI, Waist Circumference
Residual effect of muscle strength exercise in the secondary prevention of childhood obesity [39].						
To evaluate the residual effect of muscular strength exercise on body fat, metabolic syndrome, and physical fitness in schoolchildren with obesity.	This study explored the long-term effects of a 3-month muscle strength training program combined with nutritional and psychological support for obese schoolchildren in Santiago, Chile. While the program yielded positive results in terms of body fat reduction, increased walking distance, and improvement in metabolic syndrome within the 3-month period, these benefits largely faded by 9 months after the program ended. These findings suggest that long-term interventions and support systems are crucial for sustainable improvement of the health of obese children.	9	Chile	I: 111, C: 111	School setting in regular school hours	Weight, Height, BMI, Waist Circumference, lipids, and elements of metabolic syndrome
Cluster randomized trial of a school-community child health promotion and obesity prevention intervention [40].						
The community-based child obesity prevention study, fun ‘n healthy in Moreland, emerged as a research group and a local community health service with the aim of making a difference to the adverse health outcomes experienced through child disadvantage in an inner-city area of Melbourne, Australia.	“Fun ‘n healthy in Moreland!” A multiyear school-based program aimed at reducing childhood obesity in a disadvantaged area in Australia. While it did not significantly impact children’s body mass index after 3.5 years, the intervention did show success in other areas. Despite not demonstrating a direct impact on BMI, the intervention’s success in promoting healthy behaviors, policy changes, and parental engagement suggests its potential long-term benefits.	42	Australia	I: 1426, C: 1460	School setting in regular school hours, home	Weight, Height, BMI, Waist Circumference
Improving healthy dietary behaviors, nutrition knowledge, and self-efficacy among underserved school children with parent and community involvement [41].						
To measure over a 1-year period whether a CSHP with parental, school, and home-based components to promote optimal nutrition will reduce BMI percentiles and *z*-scores and improve dietary behaviors in a sample of low-income school-aged children.	A school-based program partnering with parents and the community showed promise in tackling childhood obesity among low-income Mexican American children. This one-year study enrolled 251 children in a program that combined after-school nutrition and fitness education, school-based wellness initiatives, and parent focus groups. Compared to the control group, children in the program experienced significant improvements: their BMI and BMI *z*-scores decreased; vegetable, fruit, and fruit juice intake increased; and self-efficacy for healthy food choices improved. Moreover, parental and community involvement surged to 100% by the end of the year.	12	United States of America	I: 251, C: N/A	School setting in regular school hours, school setting extracurricular, community involvement, did not include a physical activity component	Weight, Height, BMI

## Data Availability

Not applicable.

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
