# Peer review of "Health and Nutrition Interventions to Prevent Childhood Overweight and Obesity in Mexico and Latin America: A Systematic Review"

_nutrients, 2025, doi:10.3390/nu17243818_

Round 1

Reviewer 1 Report

Comments and Suggestions for Authors

Dear Authors, 

The Title of the manuscript is appropriate.

Abstract:

The abstract is well-written and addresses the most important issues raised in the manuscript.

Introduction

The Introduction is up to date and appropriate for the topic. The objectives of the work were clearly stated at the end of the introduction.

Methods: The method for finding relevant sources for the article is appropriate and has been described in detail to ensure their reproducibility. The review was conducted according to the methodology proposed by the Cochrane Collaboration [18] and the PRISMA guidelines. The "Search Strategy and Terms" section is well prepared.

Results and Discussion:

Results presented in tabular form are appropriate for presenting the findings of this review.

The structure of the review is appropriate and does not need improving. In my opinion, the work presented in this manuscript is of interest to the audience of the journal.

References:

The section on literature seems appropriate and requires a little expansion. References include articles from 2012 -2023. Why are there no articles from the last two years? Were there no articles of this type? It seems that authors should add these articles if they exist.

However, the authors do not provide any indications of new directions of research on the topic discussed.

I recommend publication of this manuscript, but the authors should add information about new directions of potential research, and try to include articles from the last two years.

Reviewer

Author Response

Comment 1. The manuscript does not incorporate the most recent evidence on childhood obesity interventions published in the last two years. Important recent trials and reviews (e.g., short-term e-health interventions, school-based social network programs, updated systematic evidence) are not discussed.

We thank the reviewer for this very helpful observation. In response, we have updated the Introduction to incorporate recent evidence published after our original search period. Specifically, in the last paragraph of the Introduction, we now summarize the updated evidence report for the US Preventive Services Task Force and three recent intervention trials

Comment 2. The Discussion does not clearly relate the present findings to the updated evidence base, nor does it sufficiently emphasize issues of equity and long-term outcomes in childhood obesity.

We agree with the reviewer that linking our findings to the most recent evidence and equity considerations strengthens the Discussion. In the Discussion, in the paragraph where we previously described Palmer et al.’s re-analysis of trials, we have now added explicit reference to the updated US Preventive Services Task Force evidence report and its implications for equity and long-term outcomes.

Comment 3. The authors do not provide clear indications of future research directions arising from their synthesis, particularly in light of newly published trials.

We appreciate this important suggestion. To address this, we have added a dedicated paragraph on future research directions in the Discussion, immediately before the paragraph on Latino/Mexican populations in the United States. This new paragraph explicitly draws on the recent trials mentioned above and connects them to the prioritiex for Latin America

Reviewer 2 Report

Comments and Suggestions for Authors

In the study entitled:  „Health and Nutrition Interventions to Prevent Childhood Overweight and Obesity: A Systematic Review”   authors aimed to  conduct a systematic review of interventions (SRI) targeting school-aged children with obesity.

Authors try to concentrate on the problem of obesity among school-children in their region, that is Mexico, a country that faces a huge obesity problem among children and adults.

The study lacks novelty and repeats in some way many existing oryginal papers, and already published meta-analyses and systematic -reviews in that field, however there is a potential for some interest for readers of Nutrients.

Overall, the study is well-written, presents good study design and description corresponding to the nature of the systematic review papers.

Remarks for authors:

Please try to  emphasise the regional potential of the paper, I would suggest to include some thougts even in the title.

Table 1 – edite in horizontal manner – for better reading.

In discussion – add some more information on the issue in question of obesity and its treatment/prevention in South America and Mexico, even if it was not the main topic of the 10 studies included in the systematic review.

Author Response

Comment 1. The regional potential of the article could be emphasized more. It would be useful to highlight, even in the title, that the findings are particularly relevant to Mexico and Latin America.

We appreciate this suggestion and agree that emphasizing the regional focus improves the clarity and relevance of the manuscript. Accordingly, we have revised the title to explicitly reflect the geographical scope:

Original title: “Health and Nutrition Interventions to Prevent Childhood Overweight and Obesity: A Systematic Review”

Revised title: “Health and Nutrition Interventions to Prevent Childhood Overweight and Obesity in Mexico and Latin America: A Systematic Review”

Comment 2. The Discussion could provide more specific information about interventions conducted in Mexico and South America to highlight the regional context and potential for application.

We agree that the regional context should be described more explicitily. In the Discussion, immediately after the paragraph describing the “Nutrition on the Go” strategy in Mexico, we have added a new paragraph that situates our findings within the broader regional experience.

Comment 3. The Conclusions could more clearly state the regional implications and potential for informing policy and practice in Mexico and Latin America.

We thank the reviewer for this suggestion. In the Conclusions section, we retained our original summary of key findings and added a final sentence that emphasizes the regional policy implications.

Comment 4. Table 1 would be easier to read if presented horizontally (landscape) to improve readability of the multiple columns.

We appreciate this practical suggestion. We have reformatted Table 1 in landscape orientation in the revised manuscript to enhance legibility and facilitate comparisions across studies. No content has been changed; only the layout has been adjusted. In the response file, we indicate the updated table as “Table 1 (reformatted in landscape orientation).”

Reviewer 3 Report

Comments and Suggestions for Authors

Although this systematic review offers important insights into childhood obesity therapies, its generalizability and rigor are limited by a number of issues. First, there are questions regarding selection bias and the representativeness of the results due to the limited scope of the included studies—just 10 of the original 127 studies were kept. It's possible that by excluding shorter interventions (less than six months), potentially successful pilot or school-based programs that offer significant behavioral insights were overlooked. Second, there is a lack of methodological transparency in the review: repeatability is weakened by the unclear description of search tactics, data extraction techniques, and inter-rater reliability for study selection. Despite the authors' use of the Mixed Methods Appraisal Tool (MMAT), readers are unable to assess the strength of the evidence base because the quality assessment results are not reported quantitatively. Furthermore, findings are limited to broad observations rather than particular evidence-based recommendations due to the heterogeneity in intervention components, settings, and outcome measures, which makes meaningful comparison or meta-analysis impossible. The ability to measure the true impact of an intervention is limited by its reliance on thematic synthesis rather than quantitative effect estimation. Lastly, socioeconomic, cultural, and environmental contexts—all of which are important factors in determining behaviors associated to obesity—are not sufficiently discussed in the study. It is unclear whether the results can be applied to a variety of groups if these contextual elements are not taken into consideration.

Author Response

Comment 1. The authors only retained 10 of 127 initially identified studies, which may introduce selection bias and limit representativeness. The exclusion of shorter interventions may have led to the omission of potentially relevant school-based pilot programs.

We thank the reviewer for this insightful comment. Our intention was to focus on multi-component community and school-based interventions with sufficient exposure time to influence medium-term anthropometric and behavioral outcomes. To clarify this rationale, we have revised the Methods section.

In the “Search Strategy and Terms” subsection, we now explicitly state that the minimum duration of six months was selected a priori and acknowledge the implications of this criterion:

“The minimum intervention duration of six months was selected a priori to focus on programs with sufficient exposure time to influence medium-term anthropometric and behavioral outcomes. As a result, shorter pilot or feasibility studies, including some school-based programs, were not retained in this review.”

In addition, in the Discussion (Limitations) we now explicitly recognize that retaining 10 of 127 initially identified articles narrows representativeness and may exclude shorter pilot programs, while improving the internal coherence of the evidence base.

We expect this makes the trade-off between focus and representativeness more transparent.

Comment 2. The review process and data extraction lack transparency. There is no information on how disagreements between reviewers were handled, nor on inter-rater reliability (e.g., kappa).

We appreciate this important methodological point. We have expanded the description of the selection and extraction processes to clarify how the discrepancies were managed.

Regarding inter-rater reliability, we did not compute formal statistics (e.g., kappa). We now acknowledge this explicitly in the “Strategy for Selection and Assessment” subsection and explain how our procedures align with the current guidance.

We also reiterate this point as part of the limitations in the Discussion.

Comment 3. The inability to perform a meaningful meta-analysis is acknowledged, but the justification remains somewhat general. The reader would benefit from a clearer explanation of why only a thematic synthesis was conducted.

We agree that the rationale for using a narrative/thematic synthesis should be more explicit. In the Results section, we originally attributed the absence of a meta-analysis to the limited The limited number of studies and their heterogeneity. We have now strengthened this justification and explicitly linked it to the choice of synthesis approach. In the paragraph describing overall findings.

In the expanded limitations paragraph of the Discussion, we further detail the types of heterogeneity that precluded a robust meta-analysis.

Comment 4. The analysis does not sufficiently address socioeconomic, cultural and environmental contexts, which are highly relevant to childhood obesity in Latin America.

We fully agree that contextual factors are crucial for understanding childhood obesity in Latin America. In the Results, we already reported how many studies considered socioeconomic status and minority groups. To make this more visible in the interpretation, we have expanded the limitations in the Discussion to explicitly address the incomplete reporting of contextual variables.

Additionally, throughout the Discussion, we have strengthened the emphasis on equity and PROGRESS characteristics by linking our findings to recent work on inequities in childhood obesity interventions (e.g., Palmer et al., 2025) and by recommending that future interventions and evaluations prospectively plan subgroup analyses by sex, socioeconomic status and other PROGRESS factors.

We believe that additions better highlight the importance of socioeconomic, cultural and environmental contexts and explicitly acknowledge the limitations of the current evidence base in this regard.

Comment 5. The strength of the evidence remains difficult to appraise. The authors mention use of the MMAT but do not present quality ratings or make them accessible. The lack of inter-rater reliability estimates further limits the reader’s ability to judge the robustness of the review.

We appreciate this constructive suggestion. As mentioned, we used the Mixed Methods Appraisal Tool (MMAT) to evaluate methodological quality, but we we did not present the results in sufficient detail. In response, we have included the following paragraph:

Two researchers independently and blindly evaluated the methodological quality of the 10 included studies using the Mixed Methods Appraisal Tool (MMAT 2018 version), re-solving discrepancies by consensus. The included studies corresponded to longitudinal, intervention, and quasi-experimental designs, all of which had moderate to high methodological quality according to the MMAT criteria. The main identified limitations were the absence of blindness for participants and evaluators (justified by the nature of the behavioural interventions in the school-age population), loss of follow-up, or difficulty in obtaining complete measurements due to deficient selection processes. These limitations were considered in the interpretation of the results and in future research recommendations.

Round 2

Reviewer 2 Report

Comments and Suggestions for Authors

Authors improved the manuscript according to my suggestions. In my opinion the article can be published in the present form. 

Author Response

We appreciate the review and comments received. 

Reviewer 3 Report

Comments and Suggestions for Authors

Nothing

Author Response

We appreciate the review and comments, the language of the manuscript has been carefully checked and corrected as suggested.